# Otological Manifestations in Adults with Primary Ciliary Dyskinesia: A Controlled Radio-Clinical Study

**DOI:** 10.3390/jcm11175163

**Published:** 2022-08-31

**Authors:** Mihaela Alexandru, Paul de Boissieu, Farida Benoudiba, Malik Moustarhfir, Sookyung Kim, Émilie Bequignon, Isabelle Honoré, Gilles Garcia, Rana Mitri-Frangieh, Marie Legendre, Bruno Crestani, Camille Taillé, Estelle Escudier, Bernard Maitre, Jean-François Papon, Jérôme Nevoux

**Affiliations:** 1ENT Department, Bicêtre Hospital, Assistance Publique-Hôpitaux de Paris (AP-HP), Paris-Saclay University, 94270 Le Kremlin-Bicêtre, France; 2Epidemiology and Public Health Department, Bicêtre Hospital, Assistance Publique-Hôpitaux de Paris (AP-HP), Paris-Saclay University, 94270 Le Kremlin-Bicêtre, France; 3Diagnostic Neuroradiology Department, Bicêtre Hospital, Assistance Publique-Hôpitaux de Paris (AP-HP), Paris-Saclay University, 94270 Le Kremlin-Bicêtre, France; 4ENT Department, Henri Mondor Hospital, Intercommuncal Hospital of Créteil, Assistance Publique-Hôpitaux de Paris (AP-HP), Paris Est University, 94010 Créteil, France; 5Mondor Institute of Biomedical Research INSERM-UPEC UMR 955, CNRS ERL7000, 94010 Créteil, France; 6Pneumology Department, Cochin Hospital, Assistance Publique-Hôpitaux de Paris (AP-HP), Paris Cité University, 75014 Paris, France; 7Pneumology Department, Bicêtre Hospital, Assistance Publique-Hôpitaux de Paris (AP-HP), Paris-Saclay University, 94270 Le Kremlin-Bicêtre, France; 8Department of Anatomy and Cytology Pathology, Intercommuncal Hospital of Créteil, 94010 Créteil, France; 9Molecular Genetics Functional Unit, Armand-Trousseau Hospital, Assistance Publique-Hôpitaux de Paris (AP-HP), Sorbonne University, 75012 Paris, France; 10INSERM, UMR-S933 Genetic Diseases of Pediatric Expression, Sorbonne University, 75005 Paris, France; 11Rare Pulmonary Diseases (FHU APOLLO) Pneumology A Department, Bichat Hospital, Assistance Publique-Hôpitaux de Paris (AP-HP), Paris Cité University, 75877 Paris, France; 12Laboratory of Excellence Inflamex, 75013 Paris, France; 13INSERM U1152, Paris Cité University, 75018 Paris, France; 14INSERM 12, F-CRIN, Clinical Research Initiative in Severe Asthma: A Lever for Innovation & Science (CRISALIS), 31059 Toulouse, France; 15Pneumology Department, Henri Mondor Hospital, Intercommuncal Hospital of Créteil, Assistance Publique-Hôpitaux de Paris (AP-HP), Paris Est University, 94010 Créteil, France; 16INSERM, U1120, Pasteur Institute, 75724 Paris, France

**Keywords:** primary ciliary dyskinesia, hearing loss, temporal bone CT scan, otitis media with effusion, chronic otitis media

## Abstract

Primary ciliary dyskinesia (PCD) is a rare genetical disease characterized by an abnormal structure or function of the cilia, causing sinusitis, otitis, and bronchiectasis. Hearing loss affects 60% of PCD patients, but data are lacking concerning hearing and temporal bone imaging in adults. Our aim was to describe clinical and radiological ear disease in adults with genetically confirmed PCD. Data were recorded from January 2018 to December 2019. PCD patients were compared with controls with bronchiectasis without PCD. Clinical examination included otomicroscopy and auditory tests. A temporal bone CT scan (TBCT) was systematically performed. Seventeen patients (34 ears) were included in each group. The eardrums were abnormal in 25 (74%) PCD ears versus 8 (24%) ears in the controls (*p* < 0.05). Conductive hearing loss was more frequent in the PCD group (24% vs. 12% in controls). TBCT were abnormal in 94% PCD patients vs. 32% in the controls (*p* < 0.05). The Main CT-scan images in PCD were middle ear inflammation (65%), mastoid condensation (62%), or ossicular anomalies (35%). With its excellent sensitivity, TBCT gives typical arguments for PCD diagnosis, adding otological signs to the usual sinus CT signs (hypoplasia, aplasia). Systematic TBCT could be useful in the initial evaluation of patients with suspicion of PCD.

## 1. Introduction

Primary ciliary dyskinesia (PCD) is a rare genetic disorder with most often an autosomal recessive inheritance. Its prevalence is around 1:2000–1:20,000 live births [1]. It is characterized by a functional (ciliary mobility) and, in most cases, the ultrastructural (e.g., dynein arms, central complex, nexin–dynein regulatory complex, cilia orientation) impairment of mucociliary clearance leading to pulmonary (chronic bronchiectasis, bronchial infections), and otological and sinonasal manifestations (chronic sinusitis and otitis media). Fertility disorders and more rarely hydrocephalus, cardiac malformations, esophageal disorders, or biliary atresia [2] may add to the clinical presentation. Situs inversus is classically reported in 50% of PCD patients and, when associated with bronchiectasis and chronic sinusitis, constitutes the Kartagener syndrome [3]. An early diagnosis improves the respiratory prognosis of PCD patients [4]. However, because of non-specific signs and non-equivocal specific diagnostic tests, PCD is very challenging to diagnose. The mean age at PCD diagnosis is 5.3 years (3.5 years for patients with a situs inversus) [5], but even nowadays, more than 5% of the patients are diagnosed during adulthood [5]. Thus, it is important to identify the specific PCD criteria in adults. Bronchiectasis is the most common pulmonary manifestation of PCD. However, this manifestation can be found in other diseases (cystic fibrosis, dysimmune diseases, severe asthma, etc.), and there is no clinical or paraclinical pulmonary sign specific to PCD. Previous ENT studies about PCD focused on sinus CT scans and showed inconsistent sinus abnormalities such as sphenoidal and frontal sinus hypoplasia [6]. It is also known that PCD patients often suffer from otological manifestations, such as otitis media with effusion (OME) or chronic otitis media (COM) (i.e., thickened eardrum, retraction pocket, and myringosclerosis) [7,8]. Consequently, hearing loss affects nearly 50% of adults with PCD [9,10]. However, to date, no study has characterized the temporal bone imaging of PCD patients or its relation with hearing function in PCD. The aim of the study was to precisely describe ear disease on CT scans in PCD and to correlate CT-scan anomalies with hearing loss in the PCD cohort of our reference center, to further identify otological criteria for PCD diagnosis.

## 2. Materials and Methods

We performed a retrospective case–control study, from January 2018 to December 2019, in consecutive adult patients with bronchiectasis referred to our PCD reference center for ciliary investigations. All the patients had bronchiectasis, a suspicion of PCD, and age >18 years on the day of ciliary testing. PCD diagnosis was considered confirmed if at least one of the following criteria was present: evidence of PCD-specific ultrastructural defects (transmission electron microscopy analysis of turbinate biopsy samples), or identification of causal mutation(s) in a known PCD gene [11]. Cases were defined as PCD patients. Controls consisted of patients with bronchiectasis and negative PCD diagnosis. Controls were matched to cases by age in a 1:1 ratio. All data were standardly recorded during routine medical care and were retrospectively analyzed. This work is part of a project with an ethical agreement (RaDiCo, IRB 00003888 dated 16 October 2015).

The following data were collected for all the patients: age, gender, history of childhood ear disease, presence of Kartagener syndrome, consanguinity in first-degree relatives, and type of ultrastructural defect. The clinical information about the patients’ sinus status was also collected (rhinosinusitis history, type of chronic rhinosinusitis, SNOT-22 quality of life score), as well as sinus CT-scan data (presence of sinus agenesis, modified Lund–Mackey score calculation). The SNOT-22 is a validated disease-specific health-related quality-of-life measure for rhinosinusitis that consists of 22 items (score range 0–110), with each item measured on an ordinal Likert scale from 0 (“no problem”) to 5 (“problem as bad as it can be”). Higher scores indicate worse symptoms [12,13], and quality of life is considered significantly impaired when the SNOT-22 score is ≥40 [14]. The modified Lund–Mackay score assesses the CT opacities of the sinus cavities, adjusted on the presence of sinus agenesis. This score ranges from 0 (no sinus opacification) to 24 (complete opacification of all sinuses) [15].

All the patients were examined by a single senior ENT physician. A bilateral and comparative otoscopy using a microscope was conducted in search of tympanic anomalies. Otitis media with effusion (OME) was characterized on otoscopy by the presence of non-purulent fluid in the middle ear. Chronic otitis media (COM) included tympanic perforation, myringosclerosis, and retraction pocket. The presence of a ventilation tube was also systematically reported.

To evaluate hearing loss, we performed an auditory test with a Madsen Astera2 audiometer (Otometrics-Natus, Massy, France). Air and bone conduction pure tone audiometry was realized with the calculation of the pure tone average (PTA) on four frequencies (0.5, 1, 2, 4 kHz) according to the modified 1995 AAO-HNS criteria [16]. Hearing loss was considered significant when the air conduction thresholds were greater than 15 dB at any frequency. Conductive hearing loss was defined as a threshold of greater than 15 dB HL and an air-bone gap of 10 dB or more. Sensorineural hearing loss (SNHL) was defined as a threshold of greater than 15 dB HL with an air-bone gap of less than 10 dB at any recorded frequency. Mixed hearing loss was defined as a threshold of greater than 15 dB HL with an air-bone gap of greater than 10 dB at the same recorded frequency [17].

Speech audiometry was also performed, using the word recognition score in quiet, and the speech reception threshold, corresponding to the decibel level at which 50% of words could be repeated by the subject. Intelligibility of 100% corresponds to the decibel level at which 100% of words could be repeated by the subject, which was also recorded for each patient.

Then, we compared the less and the more affected ear between both groups. In the PCD group, the intensity of hearing loss was also evaluated regarding the type of ultrastructure defect associated with the implicated gene.

Temporal bone CT scans (TBCTs) were performed for all the patients in the same hospital, using the same protocol, a SIEMENS Somatom Definition AS 64 (Siemens, Munich, Germany). TBCTs were obtained in helical mode, 140 kV, with mA currents between 288 and 512 mAs. The native reconstruction algorithm was a 0.6 mm slice thickness with standard filtration. All the CT scans were assessed with the Picture Archiving and Communication System (PACS) software, Carestream Health Inc (Carestream, Rochester, NY, USA). Axial images were standardized to be parallel to the lateral semicircular canal plane, and then multiplanar reconstructions were made in basic planes (axial, coronal, and sagittal). A single senior radiologist read all the TBCTs of the study, blind for the diagnosis. For each temporal bone, the radiologist evaluated the external ear (external ear canal, temporal bone), the middle ear (eardrum, middle ear, ossicular chain), the mastoid, and the inner ear.

Data are expressed as median (interquartile range) for quantitative variables, and as frequency (%) for qualitative variables. Student’s *t*-test (or Wilcoxon test, depending on the condition) was used for quantitative variables, and the chi-square test (or Fisher test) for qualitative variables to compare the differences between cases and controls. The sensitivity, specificity, positive predictive value (PPV), and negative predictive value (NPV) of the TBCTs were calculated. As PCD diagnosis was always confirmed after clinical, auditory, and radiological examinations, those data were considered “blinded” for the final diagnosis. A *p*-value < 0.05 was considered statistically significant. All analyses were carried out using Statview 4, SAS Institute, Grenoble, France.

## 3. Results

### 3.1. Patients

In total, 17 (34 ears) consecutive patients with PCD and 17 (34 ears) patients without PCD were included in this study. The median age was 39 years (IQR 33–54) in both groups, and the sex ratio was 1.33 in the PCD group and 0.75 in the control group (*p* = 0.49). The diagnosis of the disease was performed during adulthood for all PCD patients. More information about the genetic characteristics of PCD patients is reported in Appendix A.

Between both groups, gender and age did not differ significantly (Table 1). There was no heterotaxy in the control group, while three (18%) patients had Kartagener syndrome in the PCD group. There was a history of otitis media in childhood for ten (59%) patients in the PCD group and for three (18%) patients in the control group (*p* = 0.01).

The ciliary ultrastructure defects found in the PCD group were the following: the absence of inner dynein arms with microtubular disorganization for three (18%) patients, the absence of outer dynein arms for six (35%) patients, and the absence of both dynein arms for three (18%) patients. No patient had an abnormality of the central complex (CC). Five (29%) patients had normal ciliary ultrastructure.

There was a significant difference between both groups regarding their SNOT-22 score, which was higher in the PCD group (*p* = 0.01). Sinus CT scans were also significantly more pathological in the PCD group, with a higher prevalence of sinus agenesis (*p* = 0.0001) and a higher modified Lund–Mackay score (*p* = 0.01).

### 3.2. Clinical Examination

The study was performed on 68 ears. There were significantly more clinical anomalies in the PCD group: otoscopy was abnormal for 14 patients (82%) in the PCD group, and for 6 patients (35%) in the control group (*p* < 0.05) (Table 2 and Table 3).

Among the abnormal ears in the PCD group, we diagnosed OME for 13 (38%) ears and COM for 9 ears (26%). In the control group, there were three (9%) OME and five (15%) COM. No clinical signs of cholesteatoma were found.

### 3.3. Hearing Evaluation

All the patients performed auditory tests (Table 2 and Table 3). Twelve (71%) patients had hearing loss in the PCD group and eight (47%) in the control group (*p* = 0.08). Conductive hearing loss was the most frequent type of hearing loss in the PCD group (24%). In the control group, sensorineural hearing loss was found in 15% of the patients, and conductive hearing loss in 12% of the patients. The mean hearing loss was 18 dB in the PCD group and 11 dB in the control group (*p* = 0.03). Air-bone gaps were significantly higher in the PCD group, as well as the intelligibility rate at 100% (*p* < 0.05).

There was a significant difference between both groups regarding the more and the less affected ear. While the control group had normal hearing, the PCD group had significant hearing loss, even in the less affected ear (median 17.5 dB for the less affected ear and 18.5 for the more affected ear). There was no correlation between the type of ultrastructure defect and hearing loss severity in the PCD group (*p* > 0.05). There was also no significant difference in hearing loss levels between the implicated genes (*p* > 0.05).

### 3.4. Temporal Bone CT-Scan Exploration

The PCD group had 32 (94%) pathological TBCTs, while the control group had 11 (32%) pathological exams (*p* < 0.0001) (Table 4). The most frequent anomaly in the PCD group was middle ear opacification for 22 (65%) temporal bones (including tympanic membrane thickening (Figure 1) and filling out of the middle ear (Figure 2), followed by mastoid condensation for 21 (62%) (Figure 3), mastoid opacification for 13 (38%) and ossicular anomalies for 12 (35%) temporal bones (Figure 4). In the control group, eight (24%) temporal bones had mastoid opacification (*p* = 0.19) and mastoid condensation (*p* = 0.01). Seven (21%) temporal bones of this group had an inflammation of the middle ear (*p* = 0.0002), and there were three (9%) ossicular anomalies (*p* = 0.01). There was no radiological sign of cholesteatoma in either group. Inner ear malformations (SCC dehiscence or dilation of vestibular aqueduct) were found in three (9%) temporal bones in the PCD group vs. five (14%) in the control group (*p* = 0.76).

Based on these results, TBCT sensitivity was 94%, specificity was 72%, PPV was 74%, and NPV was 52%.

## 4. Discussion

To our knowledge, this is the first study reporting ear disease in adult PCD patients, including clinical and imaging explorations. It is well-established that PCD is often diagnosed late [5] because of non-specific signs. The information and training for professionals (ENT, pediatricians, pulmonologists, general practitioners, and radiologists) who are at the forefront of PCD screening is essential to improve the diagnosis performance and, therefore, the functional prognosis of these patients. Due to non-specific signs and the high incidence of ear manifestations, new clinical and paraclinical parameters are needed to improve the diagnosis of PCD, which is often delayed in adulthood.

It is known that in PCD, impaired mucociliary clearance leads to recurrent upper and lower airway infections (clinically characterized by chronic rhinorrhea and wet cough from the first months of life) and to bronchiectasis [18,19]. Our study reports significantly worse scores on SNOT-22, highlighting an alteration in the quality of life in PCD. Our results are in line with those in the literature [18].

Otological manifestations are also common in children with PCD, with a prevalence of almost 100% [8]. In this study, there was a history of otitis media (recurrent, acute, or chronic) in 59% of PCD patients (compared with 18% in the control group, *p*-value < 0.05), and more than 70% had an abnormal otological examination. Moreover, hearing thresholds were significantly higher in the PCD group. Hearing loss was present in 70% of PCD patients, with most being conductive hearing loss.

In our study, all PCD patients had a late diagnosis, during adulthood. This could partially explain the frequent conductive hearing loss found in this adult PCD population, compared with other studies (late diagnosis resulting from inadequate treatments and follow-up for OME) [9,10].

Moreover, a recent publication reported one-third of sensorineural hearing loss (SNHL) in 64 adult PCD patients, while our study reported 18%. The mean age in this PCD population was lower than that in our cohort (32 vs. 39 years), suggesting the absence of any age-related influence on hearing loss in these results [10]. More studies are needed to evaluate hearing loss more accurately in adult PCD patients and to understand the mechanisms underlying SNHL.

This is also the first study reporting TBCT explorations in PCD patients. In our study, TBCTs had excellent sensitivity. Indeed, almost all the TBCTs were abnormal, showing characteristic anomalies compared with controls. In this context, the mastoid or middle ear lesions that we described can appear secondarily to an accumulation of mucosal secretions in the Eustachian tube. This causes a lack of aeration in the middle ear, leading to chronic otitis media complications (tympanic membrane perforation, ossicular lysis, tympanosclerosis, etc.) [20]. The high proportion of ossicular anomalies found in the PCD group can also be explained by the late diagnosis resulting from years of inadequate otological treatments and follow-ups. There was more than 60% mastoid condensation in the PCD group. In line with sinus hypoplasia, this could be a congenital manifestation of PCD, and it would be interesting to study TBCTs in children with PCD to confirm it. Mastoid condensation can even occur secondarily to recurrent OME, like in the general population. Mastoid air cell opacification was found in 38% of PCD TBCTs and did not significantly differ between both groups. A recent study showed that mastoid effusion can be observed in CT scans, while patients do not have any otological manifestation [21]. However, incidental temporal bone anomalies are less probable in our study because PCD is known for its otological manifestations.

One limitation of the study is the small size of the cohort, even if it is one of the largest in such a rare disease: We explored 34 PCD and 34 control TBCTs. The retrospective pattern of our study can also introduce bias in the interpretation of the data. However, all the clinical and paraclinical exams performed in our reference center were standardized and similarly performed by a single ENT physician and radiologists having extensive experience in PCD. The PCD group was compared with a control group that had bronchiectasis but no PCD after examinations. Thus, the statistical results are more powerful when significant. Moreover, the control group was randomly matched in age with the PCD one. Hearing loss incidence is known to increase in the aging population, because of presbycusis [22]. By matching groups relative to age, we avoided the bias caused by presbycusis. Finally, we did not find a correlation between the mutated genes and the severity of hearing loss.

## 5. Conclusions

Hearing loss and temporal bone CT-scan anomalies are characteristic in adult PCD patients but non-specific. To date, no pathognomonic clinical or paraclinical symptom of the disease has been found, rather a cluster of arguments surrounding speculations on its diagnosis.

This study showed that temporal bone CT scans contribute to the ENT discourse on the diagnosis of PCD with its high sensitivity, adding new otological signs (inflammation of the middle ear, ossicular anomalies, condensation, or opacifications of mastoid air cells) to the already known and usual sinus signs (sinus hypoplasia or aplasia, opacifications). We, therefore, recommend a systematic performance of TBCTs in the initial evaluation of adult patients with bronchiectasis and suspicion of PCD. We also recommend an annual auditory evaluation of all PCD patients.

Moreover, radiologists performing temporal bone and sinus CT scans could play crucial roles in guiding clinicians toward PCD diagnosis.

These results should be confirmed by a prospective, multicenter study such as the running EPIC-PCD cohort study [23]. More studies are also needed to better understand the pathophysiology of hearing impairment in PCD.

## Figures and Tables

**Figure 1 jcm-11-05163-f001:**
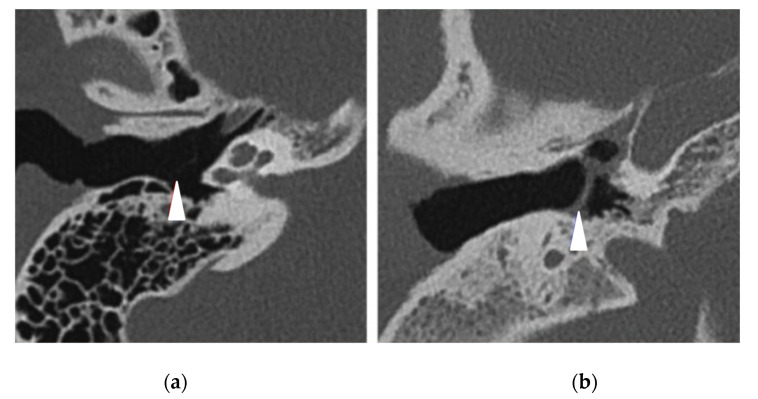
Right temporal bone CT scans in axial view showing (white arrowheads) (**a**) a normal thin tympanic membrane (control patient) and (**b**) a middle ear inflammation with a tympanic membrane thickening (PCD patient).

**Figure 2 jcm-11-05163-f002:**
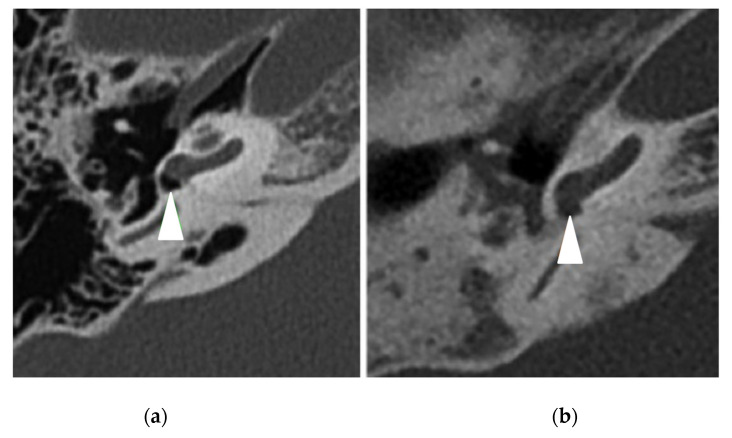
Right temporal bone CT scans in axial view showing (white arrowheads) (**a**) a normal round window (control patient) and (**b**) a filling of the round window (PCD patient).

**Figure 3 jcm-11-05163-f003:**
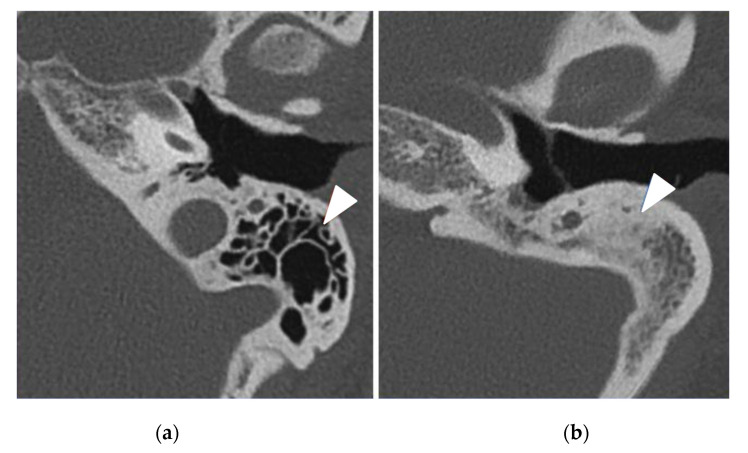
Left temporal bone CT scans in axial view showing (white arrowheads) (**a**) normal mastoid air cells (control patient) and (**b**) an important mastoid condensation (PCD patient).

**Figure 4 jcm-11-05163-f004:**
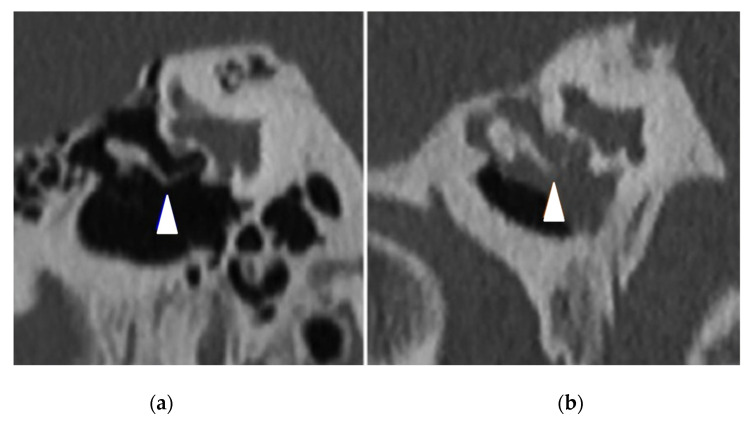
Right temporal bone CT scans in coronal view showing (white arrowheads) (**a**) a normal incudostapedial articulation (control patient) and (**b**) incus lysis associated with a chronic otitis media (PCD patient).

**Table 1 jcm-11-05163-t001:** Patients’ characteristics and otitis media history.

	PCD ^1^ Group N = 17	Control Group N = 17	*p*-Value
Age, median (Q1–Q3)	39 (33–54)	39 (33–54)	0.89
Gender male, *n* (%)	8 (47%)	6 (35%)	0.49
Ciliary ultrastructure defect, *n* (%)			
nEM ^2^	5 (29%)	17 (100%)	-
IDA + MTD ^3^	3 (18%)	0	-
ODA ^4^	6 (35%)	0	-
IDA + ODA	3 (18%)	0	-
CC ^5^	0	0	-
Kartagener syndrome, *n* (%)	3 (18%)	0	0.22
Consanguinities, *n* (%)	8 (47%)	3 (18%)	0.07
Otitis media history, *n* (%)	10 (59%)	3 (18%)	0.01
Rhinosinusitis history, *n* (%)	17 (100%)	13 (76%)	0.1
CRSwNP ^6^, *n*(%)	7 (41%)	2 (12%)	0.1
SNOT-22 ^7^ score, median (Q1–Q3)	58 (46.5–69)	33.5 (14.2–46)	0.01
Sinus hypo or aplasia, *n* (%)	15 (88%)	3 (18%)	0.0001
Modified Lund–Mackay score, median (Q1–Q3)	13.5 (10–19.2)	6 (2.7–10.2)	0.01

^1^ Primary ciliary dyskinesia; ^2^ normal electronic microscopy; ^3^ inner dynein arm defects with microtubular disorganization; ^4^ outer dynein arm defects; ^5^ central complex defects, ^6^ chronic rhinosinusitis with nasal polyps, ^7^ sinonasal outcome test-22.

**Table 2 jcm-11-05163-t002:** Otoscopy and auditory characteristics in patients’ more affected ear.

More Affected Ear	PCD ^1^ Group N = 17	Control Group N = 17	*p*-Value
Abnormal otoscopy, *n* (%)	13 (76%)	4 (24%)	0.002
− OME ^2^	7	3	
− COM ^3^	4	1	0.04
− VT ^4^	2	0	
Hearing loss, median (Q1–Q3) (dB)	18.75 (13.75–42.5)	12.5 (11.25–18.75)	0.03
Hearing loss, *n*(%)			
None	6 (35%)	12 (71%)	
Sensorineural	3 (18%)	2 (12%)	0.51
Conductive	4 (24%)	2 (12%)	
Mixed	4 (24%)	1 (6%)	
Air-bone gap 250–500 Hz, median (Q1–Q3) (dB)	5 (5–17.5)	2.5 (0–10)	0.02
Air-bone gap 1000–2000–4000 Hz, Median (Q1–Q3) (dB)	10 (2.5–12.5)	0 (0–5)	0.01
SRT ^5^, median (Q1–Q3)	25 (22.5–48.5)	23.5 (17–26)	0.06
Intelligibility 100%, median (Q1–Q3)	40 (35–65)	30 (25–40)	0.01
Hearing threshold:			
500 Hz, median (Q1–Q3)	20 (15–35)	15 (10–20)	0.04
1000 Hz, median (Q1–Q3)	20 (10–40)	10 (10–20)	0.07
2000 Hz, median (Q1–Q3)	15 (10–35)	10 (10–15)	0.08
4000 Hz, median (Q1–Q3)	35 (15–45)	20 (15–20)	0.04

^1^ Primary ciliary dyskinesia; ^2^ otitis media with effusion; ^3^ chronic otitis media; ^4^ ventilation tube; ^5^ speech recognition threshold.

**Table 3 jcm-11-05163-t003:** Otoscopy and auditory characteristics in patients’ less affected ear.

Less Affected Ear	PCD ^1^ Group N = 17	Control Group N = 17	*p*-Value
Abnormal otoscopy, *n* (%)	12 (71%)	4 (24%)	0.006
− OME ^2^	6	0	
− COM ^3^	5	2	0.01
− VT ^4^	1	2	
Hearing loss, median (Q1–Q3) (dB)	17.5 (10–33.75)	8.75 (6.25–13.75)	0.03
Type of hearing loss, *n* (%)			
None	8 (47%)	12 (71%)	
Sensorineural	3 (18%)	3 (18%)	
Conductive	4 (24%)	2 (12%)	0.49
Mixed	2 (12%)	0	
Air-bone gap 250–500 Hz, median (Q1–Q3)	5 (0–17.5)	0 (0–2.5)	0.04
Air-bone gap 1000–2000–4000 Hz, Median (Q1–Q3)	5 (0–10)	0 (0–0)	0.002
SRT ^5^, median (Q1–Q3) (dB)	25 (20–40)	22 (20–26)	0.10
Intelligibility 100%, median (Q1–Q3)	40 (30–55)	30 (25–40)	0.02
Hearing thresholds:			
500 Hz, median (Q1–Q3)	15 (10–25)	10 (10–15)	0.50
1000 Hz, median (Q1–Q3)	15 (10–20)	5 (5–10)	0.03
2000 Hz, median (Q1–Q3)	10 (10–30)	5 (5–10)	0.03
4000 Hz, median (Q1–Q3)	20 (15–35)	15 (10–15)	0.01

^1^ Primary ciliary dyskinesia; ^2^ otitis media with effusion; ^3^ chronic otitis media; ^4^ ventilation tube; ^5^ speech recognition threshold.

**Table 4 jcm-11-05163-t004:** Comparison of temporal bones CT-scan anomalies between PCD and control group.

	PCD ^1^ Group N = 34 N (%)	Control Group N = 34 N (%)	*p*-Value
Abnormal CT scan,	32 (94%)	11 (32%)	<0.0001
Mastoid opacification	13 (38%)	8 (24%)	0.19
Mastoid condensation	21 (62%)	8 (24%)	0.01
Middle ear inflammation	22 (65%)	7 (21%)	0.0002
Ossicular anomaly	12 (35%)	3 (9%)	0.01
Other:			
No other anomaly	28 (85%)	27 (79%)	
Gusher syndrome	0	2 (6%)	0.76
SCC ^2^ dehiscence	3 (9%)	3 (9%)	

^1^ Primary ciliary dyskinesia; ^2^ semi-circular canal.

## Data Availability

Not applicable.

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
