# Peer review of "Otological Manifestations in Adults with Primary Ciliary Dyskinesia: A Controlled Radio-Clinical Study"

_jcm, 2022, doi:10.3390/jcm11175163_

Round 1
Reviewer 1 Report
This review of otologic findings in PCD is important with clinical applicability for otolaryngologists, despite the small size. I have a few minor suggestions:
1. Even if the study focuses on the otologic aspect, it will be better if Table 1 includes frequencies of other findings such as sinusitis and bronchiectasis to get a full picture of PCD in the patient cohort.
2. Was any genetic study performed on the patients with PCD? If yes, please include the results and describe if prominence of otologic signs are gene- or variant-dependent.
3. A more comprehensive summary of the sinonasal and bronchial findings in previous publications in the Discussion will help. This will further highlight that the otologic aspect of PCD is understudied.
4. For the otologists who treated the described patients, did the management change upon learning that the patient(s) had PCD? If not, do the authors expect that management including otologic treatment protocols would have differed if the physicians who treated the patients had an earlier diagnosis of PCD, e.g. genetic test, knowledge of critical otologic features of PCD? Please elaborate in the Discussion.
Reviewer 2 Report
Thank your for the possibility to review your interesting paper. It deals with an important problem, which is a rather rare cause for ear problems in the overall population. This is probably the main concern: as a matter of fact TBCT-scan can clearly show alteration due to PCD, but it is not specific for it. There are more often other diseases the reasons for this findings. Your merit is to advocate attention to PCD exspecially when there are coexistent pulmonary problems.
Perhaps you can explain why you did not performed tympanometry? You should try to specify the otoscopic finding; your method seems rather rough to me.
There are some mistakes by language as far I am competent to judge this:
Page 1
5.3 years old cancel old
thickening thickende
considered confirmed which one?
Page 6
A CT can hardly prove inflammation; opacification can be found due to inflammation.
Page
th
